# Metabolic Reprogramming of Cancer Cells during Tumor Progression and Metastasis

**DOI:** 10.3390/metabo11010028

**Published:** 2021-01-02

**Authors:** Kenji Ohshima, Eiichi Morii

**Affiliations:** Department of Pathology, Osaka University Graduate School of Medicine, 2-2 Yamadaoka, Suita, Osaka 565-0871, Japan; morii@molpath.med.osaka-u.ac.jp

**Keywords:** metabolic reprograming, cancer metabolism, tumor microenvironments, anchorage-independent growth, cancer metastasis, therapeutic strategy

## Abstract

Cancer cells face various metabolic challenges during tumor progression, including growth in the nutrient-altered and oxygen-deficient microenvironment of the primary site, intravasation into vessels where anchorage-independent growth is required, and colonization of distant organs where the environment is distinct from that of the primary site. Thus, cancer cells must reprogram their metabolic state in every step of cancer progression. Metabolic reprogramming is now recognized as a hallmark of cancer cells and supports cancer growth. Elucidating the underlying mechanisms of metabolic reprogramming in cancer cells may help identifying cancer targets and treatment strategies. This review summarizes our current understanding of metabolic reprogramming during cancer progression and metastasis, including cancer cell adaptation to the tumor microenvironment, defense against oxidative stress during anchorage-independent growth in vessels, and metabolic reprogramming during metastasis.

## 1. Introduction

Cancer cells need to acquire a different metabolic state than that of non-tumor cells in order to proliferate, invade, and metastasize. During cancer progression, cancer cells encounter various kinds of metabolic stress. First, tumor microenvironments are generally hypoxic and acidic and have a distinct nutrient composition compared to non-tumor tissues from the primary site, which forces cancer cells to adapt in order to grow and invade in these environments. Second, to enter and survive in vessels, cancer cells must reprogram their metabolic state, allowing for anchorage-independent growth that induces extensive oxidative stress in cancer cells. Finally, once cancer cells colonize other organs, they must adapt to quite distinct metabolic environments than those present in primary sites [1]. Overall, because cancer cells need to reprogram their metabolic state during each step of cancer progression, metabolic reprogramming has been recognized as one of the hallmarks of cancer [2].

Elucidating the mechanisms underlying metabolic reprogramming during cancer progression can reveal the metabolic vulnerabilities of cancer cells. This may ultimately result in the identification of new therapeutic targets for cancer and improvement of patients’ prognosis. In this review, we describe each step of the metabolic reprogramming that occurs in cancer cells during cancer progression, including during growth and invasion in primary sites, survival in vessels, and colonization of other organs. Finally, we also describe emerging therapeutic strategies that target cancer-specific metabolism.

## 2. Cancer Cell Adaptation to Tumor Microenvironments

Tumor tissues exhibit an altered metabolism compared to non-tumor tissues [3,4]. Tumor metabolism is influenced by a variety of intrinsic and extrinsic factors (Figure 1) [5]. We first refer to several cell-intrinsic factors that promote tumor growth before reviewing the literature on the nutrient, oxygen, and pH statuses in tumor microenvironments.

### 2.1. Cell-Intrinsic Factors Promoting Tumor Growth

The classical example of a cell-intrinsic, reprogrammed metabolic pathway in cancer is aerobic glycolysis, which leads to the so-called “Warburg effect”, defined as an increase in the rate of glycolysis and lactate production even in the presence of oxygen [6]. This increased lactate production in turn changes extrinsic metabolic factors, including an acidic microenvironment around the cancer cells, which enhances extracellular matrix (ECM) remodeling, angiogenesis, and tumor invasion [7,8]. Oncogenes such as phosphoinositide 3-kinase (PI3K), c-MYC, and KRAS were shown to drive glycolysis by upregulating genes in the glycolytic pathway in various cancer types [5,9,10]. These oncogenes also foster glutaminolysis, an anaplerotic reaction of the tricarboxylic acid cycle (TCA) [11,12,13] that contributes to the generation of ATP in the TCA cycle and anabolic carbons for the synthesis of amino acids, nucleotides, and lipids. Therefore, glutaminolysis is considered one of the hallmarks of cancer metabolism and constitutes a potential target for cancer therapy [11,12,13]. Moreover, mitochondrial respiration and function are also required for tumor growth, although the Warburg effect is often misinterpreted as an indication that the mitochondrial oxidative metabolism is defective [3,14]. Indeed, the mitochondrial electron transport chain (ETC) is necessary for tumor growth [15,16,17] because it is coupled to the production of ATP and metabolites by the TCA cycle. Mechanistically, tumor growth requires the ETC to oxidize ubiquinol, an essential step to drive the oxidative TCA cycle [17]. Furthermore, intra-operative 13C tracing experiments in human patients revealed a prominent role of glucose oxidation in the TCA cycle in brain and lung tumors [18,19,20]. Cancer mitochondrial metabolism has thus raised as an emerging therapeutic target whose modulation has already be shown to result in anti-tumor effects [21,22].

Other cancer cell-intrinsic factors that foster tumor progression include enzyme mutations. Somatic mutations in isocitrate dehydrogenases-1 and -2 (IDH1 and IDH2), for example, occur in several tumor types, including low-grade gliomas, secondary glioblastomas [23,24], and acute myeloid leukemia [25,26]. Mutated IDH1/2 acquire a neomorphic ability to convert α-ketoglutarate (αKG) to D-2-hydroxyglutarate (D-2HG), which in turn accumulates to supraphysiological levels in IDH-mutant tumors. As a result, several αKG-dependent dioxygenases are affected, including the prolyl hydroxylases (PHDs) that degrade the hypoxia-inducible factor (HIF) alpha subunit and epigenetic modification enzymes that regulate the methylation status of histones and DNA [26]. These functions of accumulated D-2HG were proposed to promote the development and progression of tumors [26].

In addition, mutations in components of the succinate dehydrogenase (SDH) complex or in fumarate hydratase (FH) have also been extensively studied [27,28]. SDH and FH catalyze reactions in the TCA cycle, and both enzymes act as tumor suppressors in hereditary cancer syndromes [27,28]. In particular, mutations in components of the SDH complex are found in patients with hereditary paraganglioma-pheochromocytoma syndrome [29] and gastrointestinal stromal tumor [30], and mutations in FH are found in hereditary leiomyomatosis and renal cell cancer [27]. Consequently, this leads to high levels of succinate (in SDH-mutated tumors) or fumarate (in FH-mutated tumors), which interfere with the dioxygenase function of enzymes and promote tumorigenesis [29,30,31,32]. Thus, D-2HG, succinate, and fumarate share the same feature of interfering with αKG-dependent dioxygenases and are referred to as “oncometabolites”. Of note, fumarate induces epithelial-to-mesenchymal transition (EMT) through epigenetic modification in renal cancer [32]. Moreover, fumarate which could be produced by adenylosuccinate lyase, an enzyme that is involved in de novo purine synthesis, enhances cell migration capability in endometrial cancer [33].

Taken together, cancer cells alter their metabolic state due to various intrinsic factors. These include oncogenes that activate different metabolic pathways such as glycolysis and glutaminolysis, activated mitochondrial ETC function and TCA cycle in some cancer types, and mutations in enzymes such as IDH1/2, SDH, and FH.

### 2.2. Adaptation to Nutrient-Altered Tumor Microenvironments

In addition to tumor cells, the tumor microenvironment is composed of various tumor cell-surrounding cells, including fibroblasts, immune cells, and endothelial cells, as well as of acellular components such as ECM, most of which are different from those in non-tumor tissues [34]. In addition, tumor tissues have a higher cell density and more extensive fibrosis than non-tumor tissues and also a different architecture of blood and lymphatic vessels [34,35]. These characteristics lead to different nutrient availability in tumor and non-tumor tissues. Although elucidating the composition and distribution of nutrients in tumor microenvironments has been technically difficult, several studies have managed to describe this important characteristic. On the one hand, glucose levels in tumor microenvironments of several tumor types are lower, whereas lactate levels are higher, than in the plasma [36,37]. This may be due to differences in the tumor vascular architecture, which leads to reduced nutrient delivery and waste exchange between tumor cells and the circulation, or to enhanced glycolysis in tumor cells. On the other hand, a recent study found that the metabolite composition of the tumor interstitial fluid (TIF) in pancreas and lung tumors differs from that of the circulation plasma by measuring the absolute concentrations of 120 metabolites using mass spectrometry (Figure 2A) [38]. Moreover, the anatomical tumor location and dietary changes could affect the TIF composition [38].

Although it has been difficult to directly assess how altered nutrient composition in tumor microenvironments can affect cancer cell metabolism, some implications have been recently made from murine tumor models. For example, the usage of both glucose and glutamine differs between murine cells growing in culture and in vivo tumor models [38,39,40,41]. Specifically, KRAS-driven mouse lung tumors are less dependent on glutaminolysis than cultured cells, and in these tumors glucose contributes to both increased lactate production and increased TCA cycle metabolism [39]. The discrepancy in glutamine usage between the in vitro and in vivo cells can be due to the supraphysiological concentration of cystine in the culture media, which leads to the exchange between cystine and intracellular glutamate, therefore inducing glutaminolysis [40,42]. These findings indicate that the metabolic status of cancer cells can be influenced by tumor microenvironmental nutrients.

Furthermore, these findings highlight the importance of using culture media that mimics the metabolite physiological concentrations to improve the validity, reproducibility, and physiological relevance of results obtained using in vitro experiments [42]. For example, when comparing hematological tumor cells cultured in human plasma-like medium to those cultured in conventional media like Dulbecco’s modified Eagle’s medium (DMEM) and Roswell Park Memorial Institute medium, widespread effects on cellular metabolism were observed, especially an inhibition of the de novo pyrimidine synthesis caused by high concentration of uric acid [43]. Moreover, the use of a medium mimicking metabolic availability in the cerebrospinal fluid revealed the dependency of leukemic cells on stearoyl-CoA desaturase upon cell infiltration into the cerebrospinal fluid [44], and also the activation of *N*-methyl-d-aspartate receptors on breast cancer cells in brain metastatic foci [45]. Finally, in some cancer-cell types, in vitro culture using nutrient-starved medium led to the accumulation of phosphoethanolamine, which correlates with tumor growth [46].

The extracellular proteins that are present in the tumor microenvironments, including albumin, could enhance the growth of pancreatic tumors by serving as an important source of amino acids [47,48,49]. This is done through macropinocytosis, a conserved, actin-dependent endocytic process by which the extracellular fluid and its contents are internalized into cells (Figure 2B) [50]. In accordance with these observations, the inhibition of macropinocytosis suppresses the pancreatic tumor growth, indicating that macropinocytosis could be a potential therapeutic target in the treatment of pancreatic cancer [47]. Importantly, pancreatic cancer cells, but not adjacent, non-tumor pancreatic tissue cells, rely on macropinocytosis and albumin catabolism in vivo [49]. In addition, macropinocytosis of extracellular proteins, which yields amino acids including glutamine, was shown to reduce the pancreatic tumor cell dependency on extracellular glutamine [47]. Moreover, protein macropinocytosis can activate mammalian target of rapamycin complex 1 (mTORC1), which then promotes pancreatic tumor growth (Figure 2B) [48].

Taken together, the nutrient composition of the interstitial fluid in tumor microenvironments differs from that of plasma and could affect cancer-cell metabolism. Moreover, extracellular proteins serve as a source of amino acids through macropinocytosis and promote tumor growth particularly in pancreatic cancer. As a technical aspect, a medium mimicking the physiological metabolic environment can improve the in vivo relevance and reproducibility of in vitro studies, which is not the case of conventional media such as DMEM, whose composition differs from that of human physiological conditions [42].

### 2.3. Adaptation to Hypoxic Tumor Microenvironments

In contrast to normal tissues, tumor tissues are generally hypoxic environments [51,52,53,54,55]. To obtain enough oxygen to survive, cancer cells must reside within 100–200 µm of functional vasculatures, which represents the diffusion limit of oxygen in tissues [56,57]. Several regulatory systems in human cells directly sense oxygen levels and consequently influence cell physiology [58]. Among those, PHDs and the HIF pathway are the best studied examples [58,59]. PHDs are members of the αKG-dependent dioxygenase family, an oxygen-dependent enzyme. Hypoxia inhibits the activity of PHDs, whose function includes the degradation of the HIFα subunit, thus resulting in HIFα stabilization [60]. HIFs control the expression of several genes that contribute to cancer progression, including many involved in cell survival, angiogenesis, glycolysis, cancer invasion, and metastasis [58,61,62]. For example, HIF signaling induces EMT through the regulation of the EMT transcription factors SNAIL, ZEB1, and TWIST [63,64]. Moreover, HIF signaling upregulates proteolytic enzymes such as matrix metalloproteinases, which remodel the ECM [65,66] and can drive invasion and metastasis of cancer cells (Figure 3). HIFs also modulate mitochondrial function under hypoxia [58]. For example, HIF1 induces expression of lactate dehydrogenase A and pyruvate dehydrogenase kinase 1, resulting in decreased TCA cycle flux and aspartate synthesis [67,68,69]. Importantly, HIF1 also modulates the ETC function. Acute hypoxia does not inhibit ETC function, whereas prolonged hypoxia can decrease ETC function [70]. Of note, the ETC can function at near-anoxic levels [71] and, in fact, its activity was only reported to be limited in intracellular oxygen levels of 0.3% or below [72]. Indeed, pancreatic ductal adenocarcinoma (PDAC) cells require ETC activity to proliferate in severe hypoxia, with a 0.1% oxygen level [73].

αKG-dependent dioxygenases other than PHDs include DNA, RNA, and histone demethylases, such as the ten-eleven translocation DNA hydroxylases and Jumonji C domain-containing histone lysine demethylases (KDMs). For example, the histone H3 lysine 27 (H3K27) histone demethylase KDM6A and the H3K4 histone demethylase KDM5A directly sense oxygen levels and modify histone methylation marks independently of HIFs (Figure 3) [74,75]. This indicates that oxygen levels directly affect the function of chromatin regulators in modifying the cell epigenetic state. However, the role of these oxygen-regulated epigenetic modifications has not been elucidated during cancer progression and in hypoxic tumor environments, and thus requires further research.

Of note, the physiological oxygen level in normal solid organs is on average 5%, which is lower than the atmospheric oxygen level (20%), often referred to as “normoxia” [54,55]. The oxygen levels in tumors are between 0.3% and 4.2%, with most tumors showing median oxygen levels below 2% [54,55]. These indicate that the difference in oxygen concentration between tumor and non-tumor tissues may be smaller than generally considered. Therefore, culturing cells under physiological oxygen conditions as well as under pathophysiological ones is required to generate more robust in vitro experimental results and enhance the likelihood that those findings have in vivo and clinical relevance [54].

### 2.4. Adaptation to the Extracellular Acidification of Tumor Microenvironments

Extracellular acidity is a pathological feature of solid tumor tissues [76,77,78]. Acidification of the tumor microenvironment benefits tumor cell survival and growth, but not non-tumor cell survival [79,80]. Moreover, acidosis-induced adaptation also prompts the emergence of aggressive tumor cell subpopulations that exhibit increased invasion, proliferation, and drug resistance capacities [81,82,83]. Regarding the mechanisms involved in metabolic adaptation, the extracellular acidity activates the sterol regulatory element-binding protein 2 (SREBP2), which in turn induces the expression of cholesterol biosynthetic genes [84]. The expression of these genes then promotes tumor growth and correlates with decreased survival of patients with different tumors types, including glioma and pancreatic, esophageal, and breast cancers [84]. Although the lactate production in tumor cells under hypoxic conditions is thought to contribute mostly to acidification of the tumor microenvironment, acidic regions are not restricted or correlated to hypoxic areas in tumors [56,85]. Of note, the usage of a labeled pH-responsive peptide to mark acidic regions within tumors revealed that the acidic regions overlapped with highly proliferative, invasive regions at the tumor–stroma interface, which were accompanied by increased expression of matrix metalloproteinases [85].

Notably, the intracellular pH of cancer cells is higher than the extracellular pH [86,87,88]. While in non-cancer differentiated cells the intracellular pH is generally around 7.2 and thus lower than the extracellular pH of around 7.4, cancer cells have a higher intracellular pH of around 7.2 and a lower extracellular pH in the range of 6.7 to 7.1 [86]. This reversed pH gradient is recognized as one of the hallmarks of cancer and is known to enhance cancer progression and metastasis (Figure 4) [86,87,89,90,91]. In malignant pleural mesotheliomas, for example, the high intracellular pH of tumor cells specifically drives cell proliferation by inducing cAMP response element binding protein (CREB) 1–p300/CREB binding protein interaction and by promoting cyclin D1 expression [91]. High intracellular pH in cancer cells is achieved by increased activity of various plasma membrane transporters and acid efflux proteins, including monocarboxylate transporters (MCTs), Na^+^H^+^ exchangers, and carbonic anhydrases [92]. Altering the intracellular pH to an acidic condition by combination of MCT1/2 inhibition and glyceraldehyde-3-phosphate dehydrogenase (GAPDH) knockdown was shown to decrease breast cancer cell survival, indicating that interfering with the intracellular pH in cancer cells could be an effective therapeutic strategy [87].

### 2.5. Metabolic Heterogeneity in the Tumor Tissues

Tumors are metabolically heterogenous because they consist of regions with different nutrient composition, oxygen concentration, and acidity. This is not surprising considering that solid tumors have an heterogenous density of blood and lymphatic vessels and are formed by different types of fibroblasts, infiltrated immune cells, and ECM composition [93,94]. On top of these features, the distance to the vasculature is also a key factor that determines nutrient and oxygen levels in tumor microenvironments, contributing to their metabolically heterogeneity. Understanding metabolic heterogeneity is important partly because it influences the therapeutic efficacy [95]. Indeed, environmental cystine availability increases glutamine anaplerosis through Cystine/glutamate transporter (xCT)/solute carrier family (SLC) 7A11 and enhances the efficacy of glutaminase inhibition in cancer cells [40]. Moreover, environmental pyruvate levels determine the cancer cell sensitivity to metformin through modulation of the NAD+/NADH homeostasis [96].

To directly observe metabolic heterogeneity in tumors, mass spectrometry imaging (MSI) has been effectively used as a tool to visualize the spatial distribution of metabolites [97]. The combination of airflow-assisted desorption electrospray ionization–MSI, multivariable statistical metabolomics analysis, and immunohistochemistry allows a detailed study of tumor-associated metabolites and metabolic enzymes directly in their native state [98]. This spatially resolved, metabolomic combinatorial approach aids in identifying the processes that occur during cancer progression, from metabolite usage to enzyme function. This approach revealed, for example, that the functions of pyrroline-5-carboxylate reductase 2 (PYCR2) and uridine phosphorylase 1 (UPP1) are altered in esophageal squamous cell carcinoma [98]. In addition, preoperative multimodality imaging combined with intraoperative 13C-labeled glucose infusions in patients with non-small cell lung cancer revealed heterogeneity of glucose metabolism within or between tumors in relation to blood perfusion [20]. Positron emission tomography (PET) is another powerful tool that examines the metabolic state of cancer cells in vivo [99]. Recently, PET imaging of voltage-sensitive 4-[18F] fluorobenzyl-triphenylphosphonium revealed functional mitochondrial heterogeneity within individual lung tumors in autochthonous mouse models [100]. Furthermore, magnetic resonance spectroscopic imaging (MRSI) using hyperpolarized 13C-labeled metabolites has been used to examine the metabolic state of cancer cells in vivo [99]. MRSI using hyperpolarized 1-13C-pyruvate in breast cancer revealed a significant inter-tumoral and intra-tumoral metabolic heterogeneity and a correlation between lactate labeling and MCT1 expression and hypoxia [101]. Furthermore, the metabolic zonation of glioblastoma was studied by using Hoechst 33342 diffusion from vessels and by sorting viable tumor cells of an orthotopic glioblastoma cell xenograft [102]. This revealed that perivascular glioblastoma cells exhibit high protein and nucleotide synthesis and a mitochondrial content with extensive respiration governed by the restricted region of mTOR activity [102]. During collective cancer cell invasion, metabolic heterogeneity exists within invasive cell clusters, in which leader cells preferentially utilize mitochondrial respiration and trailing follower cells depend on elevated glucose uptake and on overexpression of glucose transporter 1 [103]. Targeting of leader and follower cells with pyruvate dehydrogenase and glucose transporter 1 inhibitors, respectively, inhibits cell growth and invasion [103]. Regarding the heterogeneity of enzyme expression, tumor cells at the invasive front of endometrioid carcinoma show lower expression of argininosuccinate synthetase 1, a rate-limiting enzyme in arginine biosynthesis, than other carcinoma regions [104]. The low expression of argininosuccinate synthetase 1 (ASS1) leads to decreased expression of DEP domain-containing mTOR-interacting protein, an endogenous inhibitor of mTORC1 signal, and thus enhances the activity of the mTORC1 signaling, promoting invasion of endometrial cancer cells [104].

Thus, although the technology to study metabolic heterogeneity in tumors is still limited and the imaging resolution is low, several methods including MSI, PET, and MRSI and in vivo experiments using stable isotopes or fluorescent dyes reveal intra-tumor metabolic heterogeneity. These technologies can lead to the identification of metabolic vulnerabilities that are potentially susceptible to therapeutic targeting.

## 3. Metabolic Reprograming during Anchorage-Independent Growth

To form metastases, cancer cells must detach from the ECM and enter and survive in the blood or lymphatic stream, where anchorage-independent growth is required. A very small proportion of circulating cancer cells are capable of surviving and forming metastases [105,106,107], partly because the metabolic adaptation is required in the process. On top of this, detachment from the ECM induces oxidative stress [108], and cancer cells must thus alter their metabolism to respond to this challenge in order to achieve anchorage-independent growth. To maintain redox homeostasis during anchorage-independent growth, tumor spheroids suppress oxidation of glucose and glutamine nutrients, while enhancing reductive formation of citrate from glutamine [109]. This reductive carboxylation is dependent on cytosolic IDH1 and results in the suppression of mitochondrial reactive oxygen species (ROS), maximizing anchorage-independent growth (Figure 5) [109]. Moreover, fatty acid synthase (FASN) is essential to maintain IDH1-dependent reductive carboxylation of glutamine that quenches excessive ROS produced during the transition from 2D growth to 3D anchorage-independent growth [110]. Accordingly, the genetic deletion or pharmacologic inhibition of FASN prevents tumor development and invasive growth, indicating that FASN is a potential target for cancer therapy [110]. Another factor that can enhance the survival of cancer cells after detachment from ECM is the ability of these cells to form clusters or aggregates [111,112]. Cell clustering induces hypoxia, which in turn activates mitophagy to remove damaged mitochondria and reductive carboxylation that supports glycolysis [113]. These responses reduce mitochondrial ROS production and promote metastasis [113].

On the other hand, during tumorigenesis of KRAS-mutant cells, KRAS-induced anchorage-independent growth requires mitochondrial metabolism and ROS production via regulation of the extracellular regulated kinase (ERK)/mitogen-activated protein kinase (MAPK) signaling pathway [114]. The major source of ROS generation required for this type of cell growth is the Qo site of mitochondrial complex III. The major function of glucose metabolism during KRAS-induced anchorage-independent growth is to support the pentose phosphate pathway [114]. Through gene expression analysis of sorted green fluorescent protein (GFP)-labelled breast cancer cells, circulating cancer cells were shown to exhibit enhanced mitochondria biogenesis and respiration due to upregulation of peroxisome proliferator-activated receptor γ coactivator-1 (PGC-1α) (Figure 5) [115]. Moreover, clinical analysis of human breast cancers demonstrates a significant correlation between the PGC-1α expression in invasive cancer cells and the formation of distant metastases [115]. In addition, breast cancer cells undergoing anchorage-independent growth exhibit increased mitochondrial biogenesis and upregulation of genes involved in the TCA cycle and pentose phosphate pathway (Figure 5) [116]. Overall, these gene signatures predict the potential for metastasis of breast and lung cancer cells [116].

Regarding the amino acid metabolism required for anchorage-independent tumor growth, apart from anaplerosis and reductive carboxylation of glutamine, tumor spheroids show increased alanine secretion compared to adherent cells [117]. Decreasing alanine secretion by genetical or pharmacological inhibition of mitochondrial pyruvate carrier enhanced spheroid growth and increased serine synthesis, while alanine supplementation had the opposite effect. It was thus suggested that a balance between alanine and serine may regulate spheroid viability through altered sphingolipid diversity (Figure 5) [117].

In summary, cancer cells face various stresses during anchorage-independent growth and must reprogram their metabolic state to overcome these stresses. Although the mechanism behind the metabolic reprograming depends on the cancer cell type, most examples include the reductive carboxylation of glutamine that quenches excessive mitochondrial ROS, the upregulation of pentose phosphate pathway, the enhancement of mitochondrial biogenesis and respiration, or the rewiring of a metabolic network involving serine, alanine, and pyruvate that alters sphingolipid diversity. Elucidating specific metabolic reprograming involved in anchorage-independent growth can lead to the identification of therapeutic strategies to prevent cancer metastasis.

## 4. Metabolic Reprogramming to Form Metastatic Tumors

Metastasis is a major cause of death in cancer patients. Despite various attempts to establish therapeutic strategies, few treatment strategies exist that completely cure patients with metastasis. One reason for the failure of these treatments or prevention strategies is the complex cascade involved in metastatic formation. As mentioned above, cancer cells proliferate and invade the primary tumor site and only then enter the circulation. Finally, cancer cells colonize the metastatic niche and gradually transition to an established metastasis. Cancer cells must then adapt their metabolic state to survive in the metastatic niche where the metabolic environment is distinct from that of the primary site. Elucidating the mechanisms underlying cancer cell adaptation to metastatic niches could lead to the development of new strategies against cancer metastasis. In this section, we summarize the current knowledge on the metabolic pathways and enzymes involved in enabling cancer metastasis.

### 4.1. Amino Acids

Amino acids are important for the synthesis of proteins, nucleotides, lipids, antioxidants, and TCA cycle intermediates, all of which are needed for cell proliferation and tumor metastasis. For example, endogenously produced or exogenously uptaken asparagine increases the invasive and metastatic potential of breast cancer cells and was shown to upregulate EMT genes (Figure 6A) [118]. Moreover, in primary tumors, the expression of asparagine synthetase (ASNS), which generates asparagine from aspartate, correlates with metastatic formation. Indeed, ASNS knockdown and dietary reduction of asparagine suppress breast cancer progression, indicating that ASNS can be a therapeutic target to inhibit metastasis formation in patients. Moreover, high ASNS expression is correlated with poor prognosis in many cancer types [118]. Proline catabolism via proline dehydrogenase (PRODH) also enhances growth of breast cancer cells in 3D culture conditions [119]. Moreover, PRODH expression and proline catabolism are increased in metastatic tumors compared to primary breast cancers of patients and mice. In fact, PRODH knockdown impairs the capacity of breast cancer cells to form spheroids in vitro and lung metastases in vivo. In addition, the use of the PRODH inhibitor L-THFA is sufficient to impair formation of lung metastases in orthotopic mouse models, indicating that PRODH is a potential drug target for metastasis inhibition in breast cancer patients (Figure 6A) [119]. Tumors arising from serine-limited microenvironments in mouse models of melanoma and breast cancer are reported to benefit from upregulated expression of phosphoglycerate dehydrogenase (PHGDH), a component of the serine synthesis pathway that catalyzes the rate-limiting step of glucose-derived serine synthesis [120]. The microenvironment of the brain is known to contain lower amounts of serine and glycine than the plasma [121,122]. Recently, PHGDH was shown to be a major determinant of brain metastasis in multiple human cancer types and preclinical models of breast cancer and melanoma (Figure 6A) [123]. In this particular study, genetic and pharmacological inhibition of PHGDH attenuated brain metastasis, indicating that PHGDH inhibitors may be useful in the treatment of brain metastasis [123].

### 4.2. Pyruvate, Lactate, and Acetate

Other important nutrients that support tumor metastasis are pyruvate, lactate, and acetate (Figure 6B). During cancer growth in the metastatic niche, cancer cells hydroxylate collagen to form the ECM in the metastatic niche [124]. A recent study demonstrated that pyruvate uptake induces the production of αKG and then activates collagen hydroxylation by increasing the activity of the collagen prolyl-4-hydroxylase enzyme [125]. Using two different mouse models, this study also showed that inhibition of pyruvate metabolism by pharmacological or genetic inhibition of the pyruvate transporter monocarboxylate transporter 2 is sufficient to decrease collagen hydroxylation and consequently the lung metastases of breast cancer cells [125]. Another study showed that after lung metastasis, breast cancer cells convert their metabolic state to pyruvate carboxylase-dependent refilling of the TCA cycle (anaplerosis) because of the enriched availability of pyruvate in the lung environment [126].

Lactate has been considered as a waste metabolite that is produced and exported by cells that rely on an active glycolytic system. However, some cancer cells uptake and metabolize lactate in culture [127,128]. Indeed, studies using stable isotope-labeled lactate show that mouse lung and pancreatic cancers [129] and human lung cancers [130] use MCT1 to transport lactate from the circulation into the tumor, using part of the carbon from lactate to support the TCA cycle. Of note, during the metastatic process, circulating lactate facilitates the metastatic potential of melanoma cells in patient-derived xenografts and in mouse melanomas [131]. Efficient metastatic cells have high-expression levels of MCT1, and inhibition of MCT1 reduces their lactate uptake and metastatic potential [131].

Regarding carbon sources for bioenergetics in brain metastases, metastatic mouse orthotopic brain tumors were shown to oxidize acetate in the TCA cycle [132]. Moreover, oxidation of acetate was confirmed in patients with brain metastases and was correlated with expression of acetyl-CoA synthetase enzyme 2, which converts acetate to acetyl-CoA [132].

### 4.3. Pentose Phosphate Pathway and Glutathione Synthesis

Increased activity of the pentose phosphate pathway and in glutathione synthesis support the capacity of pancreatic cancer cells to metastasize and proliferate in the liver and lung [133], and of breast cancer cells to survive in the brain microenvironment (Figure 6C) [134]. This is achieved by the effect of these two pathways in reducing oxidative stress during proliferation in the metastatic organ. As another example, during the evolution of distant metastases, PDAC co-evolved a dependence on the oxidative branch of the pentose phosphate pathway through global epigenetic reprogramming of H3K9 methylation (Figure 6C) [133]. The mechanism by which melanoma cells tend to metastasize, initially to the lymph nodes and then to other distant organs, has recently been linked to decreased oxidative stress in the lymphatic environment (Figure 6C) [135]. This may be caused by higher levels of glutathione and oleic acid and lower free-iron levels in the lymph, which decrease oxidative stress and ferroptosis. This ultimately contributes to protect melanoma cells from ferroptosis and increases their ability to form metastases in distant organs [135].

### 4.4. Lipid

Lipid metabolism plays an important role in tumor invasion and metastasis [136,137]. Cancer cells optimize their requirements for aggressive progression by switching lipid anabolism and catabolism. As for lipid anabolism, fatty acids are indispensable for the biosynthesis of most lipids, and de novo synthesis of fatty acids has been shown to drive tumor progression [138,139]. Fatty acids synthesis relies on the activation of the fatty acid biosynthetic enzymes, adenosine triphosphate citrate lyase (ACLY), acetyl-CoA carboxylase (ACC) and FASN. ACLY is responsible for de novo fatty acid synthesis as a key enzyme by converting citrate to oxaloacetate and cytosolic acetyl-CoA. Recent study showed that ACLY promotes colon cancer cell metastasis by stabilizing β-catenin 1 protein [140]. ACC is the rate-limiting enzyme in the fatty acid synthesis pathway, which carboxylates acetyl-CoA to form malonyl-CoA. High phospho-ACC expression was shown to be independently associated with worse overall survival in patients with lymph nodes metastasis of squamous cell carcinoma of the head and neck [141]. In hepatocellular carcinoma (HCC) patients, up-regulation of ACC1 was associated with multiple aggressive characteristics of HCC, such as vascular invasion and poor differentiation [142]. FASN produces saturated fatty acids, which uses one acetyl-CoA and sequentially adds seven malonyl-CoA molecules to form the 16-carbon palmitate. As an example of FASN promoting cancer metastasis, FASN expression enhanced peritoneal metastasis of ovarian cancer in part through the induction of epithelial mesenchymal transition (EMT) [143].

Fatty acids are catabolized by the fatty acid oxidation (FAO), also known as β oxidation, which sustain ATP levels and NADPH production. Thus, FAO contributes to energy production and redox homeostasis. Recent studies suggest that FAO is an important energy pathway in metastatic triple-negative breast cancer (TNBC) [144,145]. CUB-domain containing protein 1 (CDCP1) regulates lipid metabolism by decreasing cytoplasmic lipid droplet abundance, stimulating fatty acid oxidation and oxidative phosphorylation [144]. This metabolic reprogramming contributes to the energy production required for cell migration and metastasis of TNBC. The other study showed that metastatic TNBC maintained high levels of ATP through FAO and activated Src oncoprotein, which promoted TNBC metastasis [145].

Uptake of exogenous fatty acids into cancer cells could facilitate metastasis [138]. Fatty acids are translocated across the plasma membrane through either passive diffusion or a transport system. CD36 is one of the most well-characterized receptors of fatty acids in cancer cells. Recent study showed that CD36 expression in human oral carcinoma cells was essential for metastasis [146]. Mechanically, CD36-expressing metastatic cells utilize fatty acid oxidation and obtain energy required for them to survive at metastatic sites. Another study showed that adipocytes induced CD36 expression in ovarian cancer cells, which promoted metastasis of cancer cells in peritoneum and omentum [147]. And also, CD36 was shown to be a key mediator of fatty acids-induced metastasis of gastric cancer cells via the AKT/glycogen synthase kinase (GSK)-3β/β-catenin signaling pathway [148].

## 5. Emerging Therapeutic Strategies to Target Cancer Metabolism

Targeting cancer metabolism is one of the oldest chemotherapeutic strategies. In fact, it goes back to the work of Sydney Farber, who for the first time successfully used inhibitors of folate synthesis to kill leukemic cells in 1948 [149,150]. The antimetabolic chemotherapeutic reagents traditionally used in a clinical setting include L-asparaginase, the folate analogue methotrexate, the purine analogue 6-mercaptopurine, and the pyrimidine analogue 5-fluorouracil (5-FU). In recent years, the metabolic mechanisms of cancer have been substantially elucidated and various therapeutic targets have been proposed. One approach to find important metabolic pathways in specific tumors is to identify enzymes that are more expressed in tumors than in their normal tissue of origin [151,152]. Serine racemase (SRR) is a pyridoxal-5′phosphate-dependent enzyme that catalyzes the racemization of l- and d-serine, and also their dehydration to pyruvate and ammonia [153,154,155]. SRR expression is upregulated in colorectal adenoma and adenocarcinoma lesions when compared with non-neoplastic mucosa in human colorectal cancer samples [156]. SRR contributes to the proliferation of colorectal cancer cells by generating intracellular pyruvate and maintaining the mitochondrial mass. Furthermore, SRR exerts an antiapoptotic effect in these cells. Inhibition of SRR suppresses the proliferation of colorectal cancer cells and augments the efficacy of the conventional chemotherapeutic reagent 5-FU in vivo (Figure 7A) [156].

Synthetic lethality and collateral lethality are also emerging strategies to target cancer metabolism (Figure 7B) [157,158,159]. Synthetic lethality is defined as the lethal phenotype resulting from the combined loss-of-function of two or more genes, whereas the loss of each gene alone does not cause lethality. The concept of synthetic lethality has been recently applied to cancer metabolism. For example, AT-rich interactive domain-containing protein 1A (ARID1A), which is frequently mutated in ovarian clear cell and endometrioid carcinomas, was shown to maintain the glutathione homeostasis by enhancing SLC7A11 transcription [160]. Thus, ARID1A-deficient cancer cells have low basal levels of glutathione, which make these cancer cells specifically vulnerable to inhibition of the glutathione-SH (GSH) metabolic pathway (Figure 7B) [160]. Collateral lethality, on the other hand, is defined as a cancer-specific vulnerability offered by co-deletion of the genes with functionally redundant essential activities neighboring tumor suppressor genes [161]. Enolase (ENO) 1/2 in glioblastoma and malic enzyme (ME) 2/3 in pancreatic cancer are examples of genes involved in collateral lethality (Figure 7B) [161,162]. The glycolytic gene ENO1 in the 1p36 locus is deleted in approximately 5% of glioblastoma patients, which is compensated by expression of ENO2 [161]. Therefore, ENO2 inhibition selectively suppresses the growth, survival, and tumorigenic potential of ENO1-deleted glioblastoma cells [161]. Likewise, ME2 is considered a passenger deleted gene in patients with PDAC. ME2 locates in the SMAD4 locus, which is homozygously deleted in nearly one-third of the PDAC cases. ME3 depletion selectively kills ME2-null PDAC cells through impaired NADPH production [162].

Growing findings in the field of cancer metabolism keep on highlighting the importance of nutrient supply to cancer development and to treatment response. Nutritional interventions that selectively remove particular nutrients based on the mechanistic understanding of the tumor metabolic needs are an emerging concept in cancer therapy (Figure 7C) [163,164,165]. For example, dietary methionine restriction was shown to inhibit tumor growth in patient-derived xenograft models of RAS-driven colorectal cancer based on changes in one-carbon metabolism [166]. This dietary restriction augments the chemosensitivity of colorectal cancer to 5-FU in these patient-derived xenograft models and the radiosensitivity of RAS-driven autochthonous sarcoma mouse models [166]. Another study demonstrated that dietary restriction of asparagine suppressed breast cancer metastasis [118]. Other recent findings indicate that the nutritional environment affects therapy [95,167] and that the nutritional interventions increase the chemotherapeutic or radiation treatment efficacies. For example, environmental cystine increases glutamine catabolism and renders cancer cells susceptible to a glutaminase inhibitor [40]. Also, uric acid reduces the sensitivity of cancer cells to the pyrimidine analog chemotherapeutic agent 5-FU [43]. Moreover, pyruvate alters the cellular redox state, which limits the ability of metformin to suppress cancer cell proliferation [96]. Dietary supplementation of histidine increases the flux through the histidine degradation pathway and increases the sensitivity of leukemia xenografts to methotrexate [168]. Vitamins can also modulate responses to anticancer drugs. Thiamin (vitamin B1) restriction to the human physiological level in cell culture media and in an in vivo mouse xenograft model increases the response of a subset of SLC19A2-low cancer cell lines to l-asparaginase [169].

Recently, JHU083, a prodrug form of the glutamine antagonist 6-diazo-5-oxo-L-norleucine (DON), was shown to dramatically suppress tumor growth in a xenograft mouse model by decreasing the oxidative and glycolytic metabolism of cancer cells, while simultaneously boosting effector T-cell functions (Figure 7D) [170]. Of note, it was previously reported that cancer cells and T cells have the same metabolic traits, such as dependence on aerobic glycolysis and glutaminolysis [171,172]. Accordingly, a hypoxic, acidic, nutrient-depleted tumor microenvironment decreases anti-tumor immune responses [173,174,175]. This characteristic constitutes a barrier to establish novel cancer immunotherapies or therapies targeting glycolysis and glutaminolysis. The use of JHU083 may be a step forward in this regard, since it has a differential effect on cancer cells and T cells; it inhibits a broad range of glutamine-requiring enzymes including glutaminase, and consequently suppresses oxidative and glycolytic metabolism in cancer cells, resulting in decreased hypoxia, acidosis, and nutrient depletion [170]. Furthermore, in effector T cells, JHU083 markedly upregulates oxidative metabolism and promotes a long-lived, highly-activated phenotype [170]. This study opens the possibility to differentially modulate the metabolism of cancer cells and anti-tumor immune cells, which can synergistically enhance the anti-tumor effects [170,176].

Taken together, although some therapeutic reagents targeting cancer metabolism, like methotrexate and 5-FU, have been used for a long time, growing new findings in the field of cancer metabolism have led to the development of various emerging strategies. Although most of these emerging strategies are still in the preclinical stage, further research is expected to bring these new strategies to the existing clinical settings.

## 6. Concluding Remarks

Numerous studies have elucidated the mechanisms underlying metabolic reprogramming of cancer cells during cancer progression. These studies also revealed the metabolic vulnerabilities of cancer cells, which could lead to the development of new therapeutic strategies targeting cancer cell metabolism. However, the complete picture of cancer metabolism has not been fully unraveled because metabolic reprogramming of cancer cells is an extremely complex process influenced by cell intrinsic genetic determinants and numerous environmental factors such as non-tumor cells, density of vessels, acellular components such as ECM, and the metabolic state of patients. Furthermore, several discrepancies between in vitro and in vivo findings still exist. Developing new technologies, such as high-resolution in vivo metabolic imaging, and improving experimental methods, such as the use of culture systems with physiological nutrient and oxygen levels, will reveal more complex mechanisms underlying metabolic reprogramming of cancer cells. These improvements are expected to bring a therapeutically relevant view to the cancer metabolism research.

## Figures and Tables

**Figure 1 metabolites-11-00028-f001:**
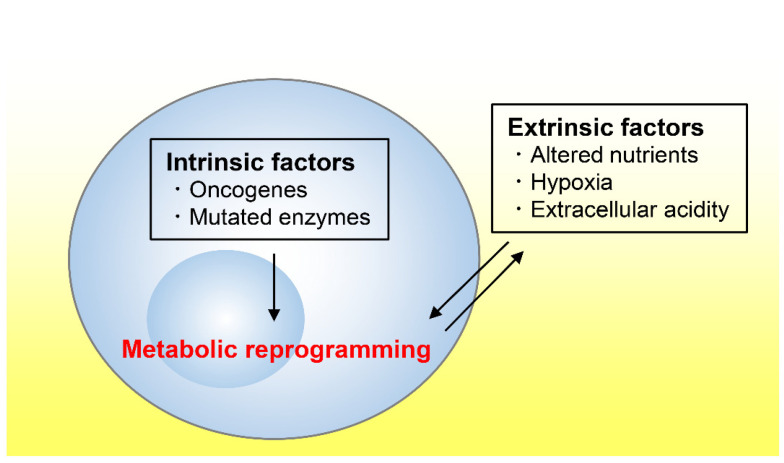
Schematic view of metabolic reprograming of cancer cells. Both intrinsic and extrinsic factors induce metabolic reprograming of cancer cells. Intrinsic factors include oncogene and mutated enzymes, and extrinsic factors include altered nutrients, hypoxia, and extracellular acidity in tumor microenvironments.

**Figure 2 metabolites-11-00028-f002:**
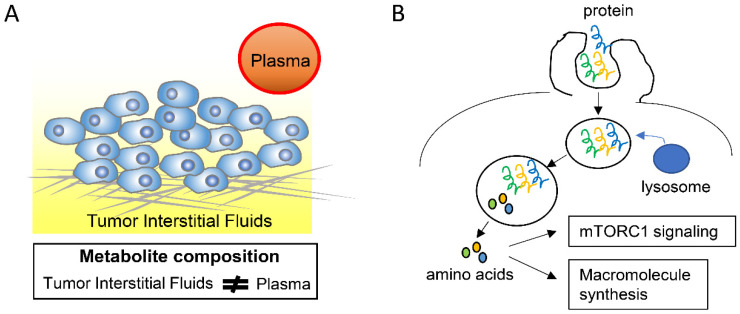
Adaptation to nutrient-altered tumor microenvironments. (**A**). Measuring the absolute concentrations of 120 metabolites with mass spectrometry identified that the metabolite composition of the tumor interstitial fluid in pancreas and lung tumors differs from that of the circulation plasma [38]. (**B**). Macropinocytosis of extracellular proteins serves a source of amino acids in pancreatic cancer cells, which could activate mechanistic target of rapamycin complex 1 (mTORC1) signaling and be used for macromolecule synthesis.

**Figure 3 metabolites-11-00028-f003:**
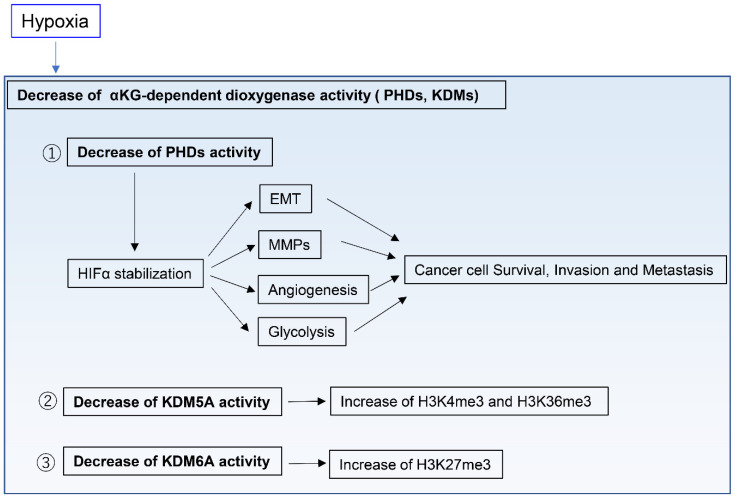
Adaptation to hypoxic tumor microenvironments. Hypoxia decreases αKG-dependent dioxygenase activity such as prolyl hydroxylases (PHDs) and histone lysine demethylases (KDMs), which affect cancer cell property and histone H3 methylation status. HIF; hypoxia-inducible factor, EMT; epithelial-to-mesenchymal transition, MMPs; Matrix metalloproteinases.

**Figure 4 metabolites-11-00028-f004:**
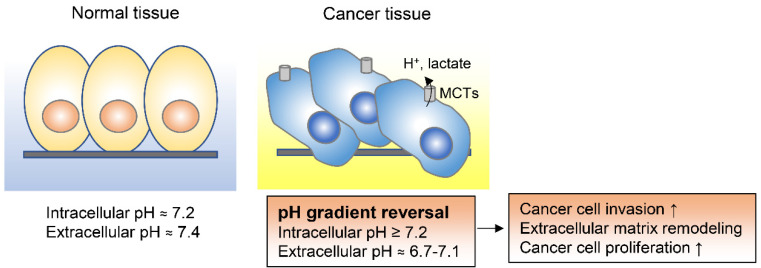
Adaptation to the extracellular acidification of tumor microenvironments. Cancer cells have reversed pH gradient compared to normal tissue which is achieved by monocarboxylate transporters (MCTs). The reversed pH gradient could enhance cancer cell invasion and proliferation.

**Figure 5 metabolites-11-00028-f005:**
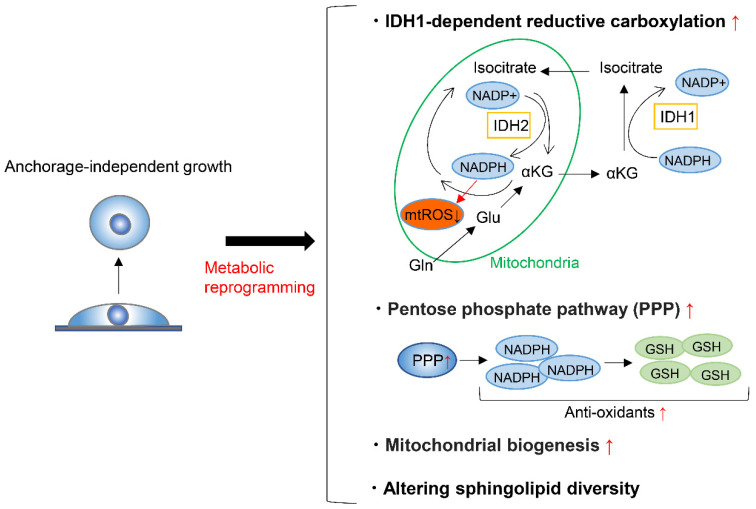
Metabolic reprograming during anchorage-independent growth. Metabolic reprograming during anchorage-independent growth include isocitrate dehydrogenases-1 (IDH1)-dependent reductive carboxylation of glutamine, upregulation of pentose phosphate pathway, the enhancement of mitochondrial biogenesis and respiration, and altering sphingolipid diversity. mtROS; mitochondrial reactive oxygen species, GSH; glutathione-SH.

**Figure 6 metabolites-11-00028-f006:**
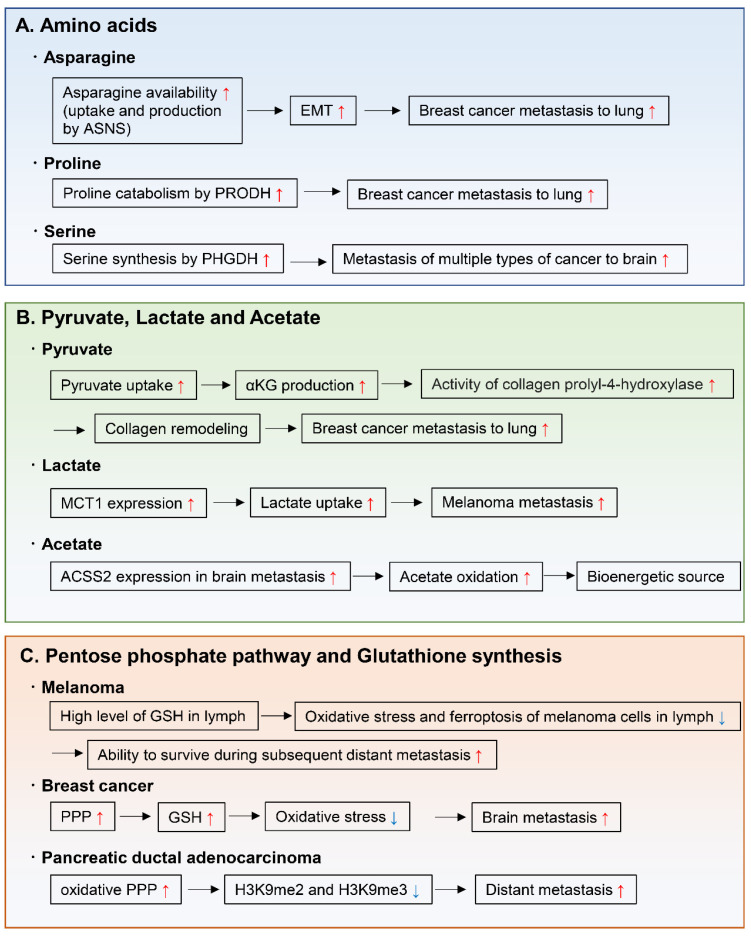
Metabolic reprogramming to form metastatic tumors. The metabolic pathways and enzymes involved in enabling cancer metastasis include those of (**A**) amino acids, (**B**) pyruvate, lactate and acetate, and (**C**) pentose phosphate pathway and glutathione synthesis. ASNS; asparagine synthetase, EMT; epithelial-to-mesenchymal transition, PRODH; proline dehydrogenase, PHGDH; phosphoglycerate dehydrogenase, αKG; α-ketoglutarate, MCT1; monocarboxylate transporter 1, ACSS2; acetyl-CoA synthetase 2, PPP; pentose phosphate pathway, GSH; glutathione-SH.

**Figure 7 metabolites-11-00028-f007:**
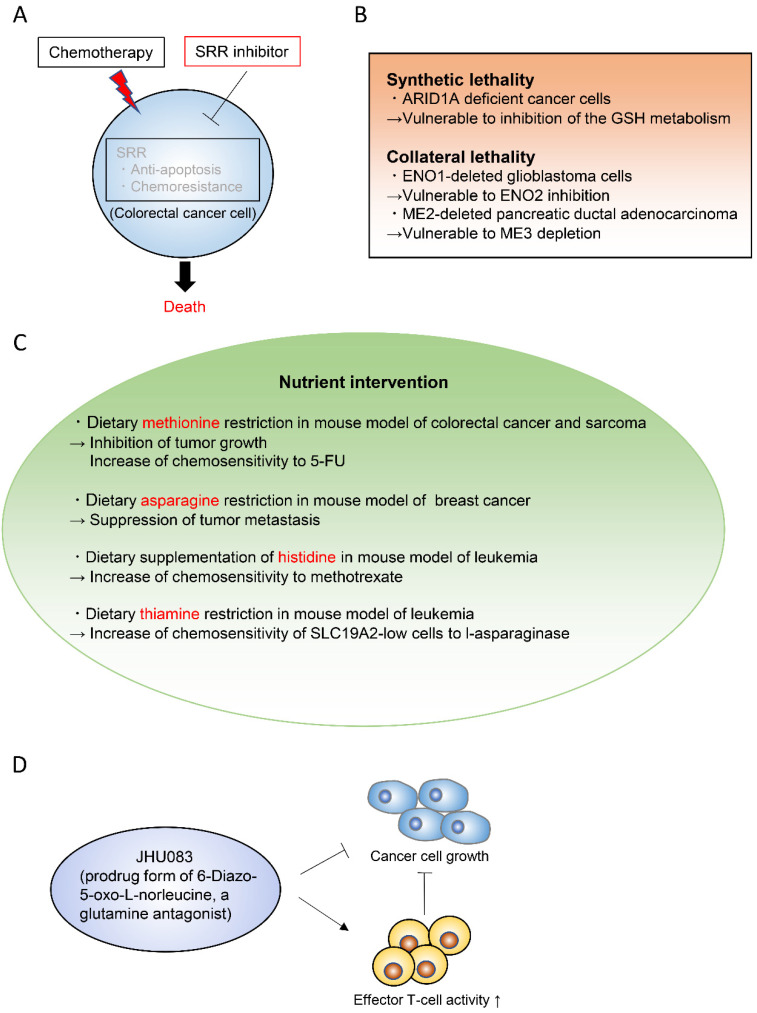
Emerging therapeutic strategies to target cancer metabolism. (**A**). Inhibition of serine racemase (SRR) augments the efficacy of chemotherapy [156]. (**B**). Synthetic lethality and collateral lethality are emerging strategies to target cancer metabolism. (**C**). Nutritional interventions that selectively remove or supply particular nutrients could suppress tumor growth or enhance chemosensitivity. (**D**). JHU083 differentially modulate the metabolism of cancer cells and anti-tumor immune cells, which can synergistically enhance the anti-tumor effects [170]. SRR; serine racemase, ARID1A; AT-rich interactive domain-containing protein 1A, GSH; glutathione-SH, ENO; Enolase, ME; malic enzyme, 5-FU; 5-fluorouracil, SLC; solute carrier.

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
