# Peer review of "Metabolic Reprogramming of Cancer Cells during Tumor Progression and Metastasis"

_metabolites, 2021, doi:10.3390/metabo11010028_

Round 1

Reviewer 1 Report

Cancer dependent metabolic rewiring is one of the hallmarks of cancers. In this review, the authors describe the role of metabolic processes in cancer cell adaptation to the tumor microenvironment, anchorage-independent growth, and metastatic tumors. Finally, they represent emerging therapeutic strategies to target cancer metabolism. I found this review to be thorough, with only a few minor comments.

Minor comments

  1. Row 160, the “A” needs to be bold.
  2. Add the gene symbol to the full name, row 214, row 234, row 260, two enzymes, and row 283.
  3. Change typo in row 424

Reviewer 2 Report

Ohshima and Morii have compiled a very well written and informative review about metabolic reprogramming in cancer cells. The authors summarized cell intrinsic and extrinsic factors influencing metabolism, and furthermore, reviewed the literature about metabolic changes during migration and invasion processes in cancer cells. Finally, strategies looking at targeting the particular metabolic features of cancers are considered.

I would like to bring forward a few minor points for the authors to consider for revision.

  1. In the abstract, introduction and section 2.2 (Figure 2) the phrase “nutrient-altered” or “altered nutrient composition” is used. After reading section 2.2 I understand what the authors aim to express, I feel these phrases do not really do though, since it is not entirely clear in what way nutrients are altered. Maybe phrases like “tumor-specific nutrient availability” or “distinct nutrient composition compared to non-cancerous tissue from the primary site” might be clearer to the reader.
  2. Line 258 rephrase, since MSI itself captures snapshots of metabolic patterns in time: “… combinational approach aids in identifying processes …”
  3. Under point 4, metabolic reprogramming to from metastatic tumors, the authors should briefly review the role of glycolysis versus oxidative phosphorylation and lipid metabolism for migration processes and EMT.
  4. The review contains multiple examples for tissue-of-origin specific or gene mutation related metabolic dependencies that were described in the literature. For this reason, I would add to the concluding remarks, line 519-522, “… influenced by cell intrinsic genetic determinants and numerous environmental factors …”
  5. Typos: Figure 3 (Cacer cell Survival), Line 225 (pH of, not than), line 329 (space missing), line 372 (use)
  6. English corrections: Line 161/162 consider rephrasing, line 239 (reversed pH gradients compared to normal tissue …)
